# Searching Scientific Literature for Answers on COVID-19 Questions

**Vincent Nguyen**[1,2]   **Maciej Rybinski**[1]   **Sarvnaz Karimi**[1]   **Zhenchang Xing**[2]

[1]CSIRO Data61, Sydney, Australia

[2]The Australian National University, Canberra, Australia

{firstname.lastname}@csiro.au

{zhenchang.xing}@anu.edu.au

## Abstract

Finding answers related to a pandemic of a novel disease raises new challenges for information seeking and retrieval, as the new information becomes available gradually. TREC COVID search track aims to assist in creating search tools to aid scientists, clinicians, policy makers and others with similar information needs in finding reliable answers from the scientific literature. We experiment with different ranking algorithms as part of our participation in this challenge. We propose a novel method for neural retrieval, and demonstrate its effectiveness on the TREC COVID search.

## 1 Introduction

As COVID-19—an infectious disease caused by a coronavirus—led the world to a pandemic, a large number of scientific articles appeared in journals and other venues. In a span of five months, PubMed alone indexed over 46,000 articles matching coronavirus related search terms such as `SARS-CoV-2` or `COVID-19`. This volume of published material can be overwhelming. There is, therefore, a need for effective search algorithms to help in finding the relevant information, and for question-answering systems able to suggest the correct answers to a given information need. In response to this need, an international challenge—*TREC COVID Search* (Roberts et al., 2020)—is organised by several institutions, such as NIST and Allen Institute for AI, where research groups and tech companies develop systems that search over scientific literature on coronavirus. Through an *iterative* setup organised in different rounds, participants are presented with several topics. The evaluations measure the effectiveness of these systems in finding the relevant articles containing answers to the questions in the topics.

We detail our participation in the challenge and propose a novel neural ranking approach. Neural rankers are mostly used for reranking a set of retrieved documents, as the cost of neural retrieval over the entire collection is high (Sebastian Hofstatter, 2019). It is also shown that semantic neural models, such as BERT (Devlin et al., 2019), do not produce universal sentence embeddings (Reimers and Gurevych, 2019). To alleviate the shortcomings of using a neural index, we propose a hybrid index with both an inverted and neural index for ranking. Our fully automatic model scores highly in the TREC COVID challenge with minimal tuning or task-specific training. We compare our neural index with strong baselines and the method yields promising results.

## 2 Related Work

The use of neural networks in search has mostly been limited to reranking top results retrieved by a 'traditional' ranking mechanism, such as Okapi BM25. Only a portion of top results is rescored with a neural architecture (McDonald et al., 2018). Since the most successful neural reranking models depend on joint modelling of both documents and the query, rescoring the entire collection becomes costly. Moreover, the effectiveness gains achieved with neural reranking are debated (Yang et al., 2019) until recently (Lin, 2019).

Since late 2018, large neural models pre-trained on language modeling—specifically BERT which uses bi-directional transformer architecture—achieves state-of-the-art for several NLP tasks. The architecture is then successfully applied to ad-hoc reranking (Nogueira and Cho, 2019; Akkalyoncu Yilmaz et al., 2019; Dai and Callan, 2019).

The existing applications of BERT in search share the limitation of being restricted to reranking, because they rely on its next sentence prediction mechanism for a regression score. In contrast, our approach builds on (Reimers and Gurevych, 2019), where a BERT architecture is trained to produce sentence embeddings. Leveraging these

```
<topic number="3">
  <query>coronavirus immunity</query>
  <question>
    will SARS–CoV2 infected people develop
    immunity? Is cross protection possible?
  </question>
  <narrative>
    seeking studies of immunity developed
    due to  infection with SARS–CoV2 or
    cross protection gained due to
    infection with other coronavirus types
  </narrative>
</topic>
```

Figure 1: A sample topic from the COVID search task.

embeddings allows for a cost-efficient application of BERT to neural indexing.

## 3  Dataset

**Documents**  CORD-19 (The Covid-19 Open Research Dataset) (Wang et al., 2020) is a dataset of research articles on coronaviruses (COVID-19, SARS and MERS). It is compiled from three sources: PubMed Central (PMC), articles by the WHO, and bioRxiv and medRxiv. The collection grew to over 68,000 articles by mid-June 2020. The growth of CORD-19 continues with weekly updates (Roberts et al., 2020).

**Topics**  As part of TREC COVID search challenge, NIST provides a set of important COVID-related topics. Over multiple rounds, the topic set is augmented. Round 1 has 30 topics, with five new topics added per subsequent round. A sample topic is shown in Figure 1. Each topic consists of three parts: query, question, and narrative.

**Relevance Judgements**  The nature of the COVID search requires an ongoing manual review of search results for their relevancy. TREC organises manual judgements per each round, using a pooling method over a sample of the submitted runs. Given a topic, a document is judged as: irrelevant (0), partially relevant (1), and relevant (2).

## 4  Methods

Details of some of our (CSIROmed team) promising approaches are explained below.

### 4.1  Round 1

**NIR: Neural Index Run**  We build a neural index (Zamani et al., 2018) by appending neural representation vectors to document representations of a traditional inverted index. The neural representations were created using the pooled classification token, *[CLS]*, from the BioBERT-NLI model[1] to produce universal sentence embeddings (Conneau et al., 2017) for the title, abstract and full-text facets. The BioBERT-NLI model is derived from applying the training of the Sentence Transformer (Reimers and Gurevych, 2019), a Siamese network built for comparison between transformer sentence embeddings, to the BioBERT pretrained model (Lee et al., 2019). To obtain sentences, we use sentence segmentation libraries. For Round 1, a rule-based approach *segtok*[2] is applied. We use a neural approach, *ScispaCy* (Neumann et al., 2019), for all subsequent runs.  We produce sentence-level representations by passing the individual sentences through the model.  We then produce and index field-level representations by averaging over sentence-level representations.

For all rounds, we indexed the collection with a single V100 Nvidia GPU with 8 CPU cores and 64 GB RAM at a rate of 2100 documents per hour using Elasticsearch for our index and bert-as-service[3] for fast embeddings.

For retrieval, we introduce a hybrid approach. We score topic-document pairs by combining: (i) Okapi BM25 scores for all pairs of topic fields and document facets; and (ii), cosine similarities calculated for neural representations of all pairs of topic fields (calculated ad hoc) and document facets (stored in the index). The final score is obtained by adding a log-normalised sum of BM25 scores (i) to the sum of neural scores. More formally, the relevance score $\psi$ for i$^{th}$ topic $T_i$ and document $d \in D$ is given by:

$$\psi(T_i, d) = \log_z(\sum_{}^{t \in T_i} \sum_{}^{f \in d} BM25(t, f)) \\ + \sum_{}^{t \in T_i} \sum_{}^{f \in d} cos(v(t), v(f)), \quad (1)$$

where z is a hyper-parameter, $t \in T_i$ represents possible fields of the topic (i.e., query, narrative and question), $f \in d$ represents possible facets of the document (i.e., abstract, title, body), BM25 denotes the BM25 scoring function, $v(t)$ denotes the neural representation of the topic field, $v(f)$ denotes the neural representation of the document facet, and *cos* denotes cosine similarity. The hyper-parameter

---

[1] https://rb.gy/toznrv (Last Accessed: 23/6/20)
[2] https://rb.gy/ashfot (Last Accessed: 19/4/20)
[3] https://rb.gy/wlevnc

$z$ is set such that the highest scoring document has a value of nine. We also filter by date, documents created before December 31st 2019 (before the first reported case) are removed.

We use a hybrid neural index as the model (Reimers and Gurevych, 2019) has not been pretrained with a ranking objective. We release our code for the neural index in GitHub.[4]

**RF: Relevance feedback baseline**   We indexed the collection using Apache Solr 8.2.0 with the following fields: abstract, title, fulltext, date. We used the default Solr setting for preprocessing.

For ranking, we created a relevance model (RM3) for query reformulation using all fields of the original topics and a subset of human-judged relevant documents.[5]

For search with the reformulated queries, we used divergence from randomness (DFR) similarity (I(n) is inverse document frequency model with Laplacian after-effect and H2 normalisation with $c = 1$). For each topic, we used expanded queries of up to 50 terms, with the interpolation coefficient 0.4 with the original query terms. Filtering by date was also applied.

### 4.2   Round 2 and 3

We produce two neural index runs for Round 2 and 3, a run (NIR) with the same parameters as Round 1, however, we use *ScispaCy* for sentence segmentation. Our second run, NIRR, is similar to NIR run but additionally reranks using the top three sentence scores from abstract and fulltext sentence vectors similar to Akkalyoncu Yilmaz et al. (2019). We fuse NIR and NIRR runs for Round 3.

**RFRR: Relevance feedback with BERT-based re-ranking baseline**   We used indexing from relevance feedback baseline from Round 1. We also used DFR retrieval with relevance feedback query expansion (using Round 1 judgements). Neural re-ranking of top 50 results was done with a SciBERT model with additional pretraining on target corpus[6] and fine-tuned on a wide variety of biomedical TREC tasks: TREC Genomics 2004-2005, TREC CDS 2014-2016, TREC PM 2017-2019 and COVID-TREC Round 1 judgements. Fine-tuning was carried out as a binary classification training using BERT next sentence prediction, with query

[4] https://rb.gy/5ahwm2
[5] https://rb.gy/egpvgd (Last Accessed: 1/7/20)
[6] https://rb.gy/mkfifz (Last accessed: 13/5/20)

| Round | Run | NDCG@10 | P@5 | MAP | Bpref |
|-------|-----|---------|-----|-----|-------|
| **1** | NIR$_{[cls]}$ | 0.588 | 0.660 | 0.217 | 0.407 |
|       | RF | 0.548 | 0.640 | 0.245 | 0.418 |
|       | Best | 0.608 | 0.780 | 0.313 | 0.483 |
| **2** | NIR$_{[cls]}$ | 0.578 | 0.720 | 0.192 | 0.376 |
|       | NIRR | 0.568 | 0.703 | 0.186 | 0.371 |
|       | RF-RR | 0.580 | 0.680 | 0.218 | 0.436 |
|       | Best | 0.625 | 0.749 | 0.284 | 0.679 |
| **3** | NIR$_{[cls]}$ | 0.625 | 0.725 | 0.1913 | 0.418 |
|       | NIRR | 0.488 | 0.680 | 0.141 | 0.389 |
|       | Fusion | 0.493 | 0.575 | 0.144 | 0.395 |
|       | Best | 0.687 | 0.780 | 0.316 | 0.566 |

Table 1: Effectiveness of our official submitted runs. Best represents the highest automatic run's scores from the TREC leaderboard[7].

presented to the model as text A and abstract or other abbreviated document representation (i.e., summaries of clinical trials for TREC PM collections) presented to the model as text B.

## 5   Evaluation and Results

**Metrics**   Three precision focused metrics are used to evaluate the effectiveness of the rankings: NDCG at rank 10 (NDCG@10), precision at rank 5 (P@5), mean average precision (MAP). BPref is reported as a metric that takes into account the noisy and incomplete judgements in this task.

**Results**   We present two sets of evaluations: (1) results of our runs submitted for the challenge, which we call *official* runs in Table 1; and, (2) results of additional runs which are not submitted but evaluated using the released relevance judgements in Table 2.

The results in Table 1 show that the NIR model outperforms a strong baseline (RF) and performs on par with the relevance feedback run with neural reranking, without the need for task specific training data.

## 6   Insights

**Ablations**   From Table 3, we deduce that the neural component has no inherent concept of relevance (semantic equivalence, does not imply relevance). This is shown when ranking (NDCG@10) improved when removing cosine scores from the relevance calculation. This effect is likely from the retrieval of unjudged documents. However, this could be remedied by using a stronger model for the task (ClinicalCovid-NLI, Table 2). We also experimented with the BioBERT-msmarco model

[7] https://rb.gy/plcuhv

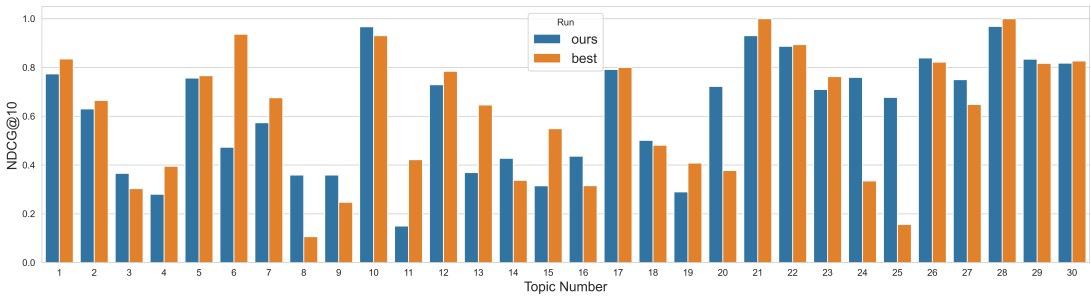

Figure 2: Comparison of NDCG@10 scores between our $\text{NIR}_{avg}$ and the top automatic run for every topic

| Model | Round 1 | Round 2 |
|---|---|---|
| $\text{NIR}_{avg}$ (baseline) | 0.614 | 0.608 |
| Covid-NLI | 0.582 | 0.522 |
| ClinicalCovid-NLI | **0.641** | **0.650** |
| BioBERT 1.1 STS | 0.612 | 0.613 |
| BioBERT-msmarco | 0.232 | 0.593 |
| SciBERT-NLI | 0.594 | 0.570 |
| Best Automatic Run | 0.608 | 0.625 |

Table 2: NDCG@10 for our additional runs for Round 1 and 2. Best run is as reported by organisers per that round. For $\text{NIR}_{avg}$, average pooling is used instead of classification token pooling ($\text{NIR}_{[cls]}$).

| Model | P@5 | NDCG@10 |
|---|---|---|
| $\text{NIR}_{avg}$ | 0.747 | 0.615 |
| - neural | 0.700 | 0.624 |
| - bm25 | 0.307 | 0.277 |
| - title | 0.700 | 0.592 |
| - abstract | 0.180 | 0.123 |
| - fulltext | 0.733 | 0.644 |
| - filter | 0.707 | 0.589 |

Table 3: Ablation study on each document facet using primary ranking metrics from Round 1.

that was heavily used by top teams for re-ranking, but had poor results for neural indexing.

**Comparison with Top Run** From Figure 2, we found some anomalies in the graph. Our model performs better on Topics 24–25 than the best automatic run. One reason is that the documents retrieved by the best run had a significant vocabulary overlap with the query. For instance, the most relevant document for Topic 25, was about complications related to diabetes generally and not complications in diabetic patients with coronavirus. Our neural component seems to alleviate this effect of pure word-based matching.

**Where does the model succeed or fail?** There is some bias in the TREC judged documents. Most of the participants' runs were created using a word-matching algorithm, causing our neural index to have many unjudged documents in the top 10. For example, for Topic 3 (Figure 1) the document at rank 3 for shared no keywords with the query, "coronavirus cross immunity", but had "antibody-dependent enhancement from coronvarius-like diseases". This phrase is highly relevant to the query, but the document is unjudged.

We found that our model placed an irrelevant document at the rank one for Topic 3. This document was scored highly by BM25 but much lower in the neural/cosine component. It was scored highly as it repeated many of the keywords in the query, however, the semantic content of the text was irrelevant to the query itself as it discussed "coronavirus crossing continents".

We expect the top rank document to be scored highly by both components; however, we assumed that the score range of the neural retrieval would have an upper limit of 9. However, this could only occur if the title, abstract and full text were all equally similar to the topic, which is unlikely; this makes the scorer biased to BM25. In contrast, looking at the second document retrieved, we found that BM25 placed this document outside the top 10. However, since the neural model scored it highly, it was moved to the second position.

## 7 Conclusions

We propose a novel neural ranking approach, $\text{NIR}_{avg}$, that is competitive compared to other automatic runs in the COVID TREC challenge. We show that a neural ranking is beneficial, but has some drawbacks, which may be alleviated when paired with a traditional inverted index. Learning a balanced scoring function to combine the strengths of the inverted and neural indices or using only a neural index explicitly trained for ranking would be a suitable avenue for future research.

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
