# OpenReview forum: "Searching Scientific Literature for Answers on COVID-19 Questions"
_EMNLP/2020/Workshop/NLP-COVID — Submitted to NLP-COVID19-EMNLP_

### Official Review · AnonReviewer4 · 2020-09-24
**Well presented summary and extended experiments for robust TREC-COVID challenge system**

**Rating:** 8
**Confidence:** 4

**Review:**

This article describes IR methods and their evaluation on the context of the TREC-COVID challenge. The article is well written and structured, and the authors do a good job in providing the full context of the challenge, together with their own contributions. Given the space constraints, I believe the information is very well summarised.

With regards to the technical contributions, the methodology they applied to build and evaluate their models is sound, and the error analysis is insightful, providing useful ideas for future work. Maybe it would have been interesting to provide more details about the systems in Table 2. For instance, the different NLI and BioBERT versions should be explained, and also the difference between NIR_avg and NIR_cls (not just in the table caption). I think it would be more relevant to have a paragraph covering the Table2 variations than keeping Figure 2 (the example is interesting, but the figure takes too much space in my opinion).

Minor comments:
- NIR_avg has slightly different numbers for Round 1 NDCG@10 in Table 2 (0.614) and Table 3 (0.615)
- P1: "gains achieved with neural reranking are debated until recently": a bit of elaboration would help provide more context.
- P1: "large neural models pre-trained on language modeling—specifically BERT which uses bi-directional transformer architecture— achieves": should be "achieve"
- P4: "the document at rank 3 for shared no keywords": remove "for"

---

> ### Author Response · Authors · 2020-09-27
> **Differences between models and avg/cls pooling methods**
>
> Thank you for your review, I wanted to address some concerns you've raised which we can add this to the final version of the paper as well.
>
> **"...the different NLI and BioBERT versions should be explained..."**
>
> The main differences in table 2 between the models is that the base models are finetuned on either NLI/STS datasets (using Sentence-BERT method to produce universal sentence embeddings) or the ms-marco (passage re-ranking dataset for the model to learn to rank). The base models are BioBERT, CovidBERT, SciBERT and ClinicalCovidBERT.
>
> **"...the difference between NIR_avg and NIR_cls..."**
>
> When inspecting the model's final attention layer, we noticed that the models did not select [CLS] as a global token to pool information into which meant we lost information/performance. We then found that taking an average of the final layer yielded better results. Rounds 1 - 3, used [cls] pooling (just taking the classification embedding) overtaking an average of all outputs in the final layer.

---

### Official Review · AnonReviewer2 · 2020-09-25
**[revised - lean accept assuming revisions] Detailed report of participating system in TREC challenge; moderate performance; few learnings/insights**

**Rating:** 6
**Confidence:** 4

**Review:**

Final update:  Per continued dialogue with the authors (see thread below), I've modified my rating of this work from 4 to 6, leaning toward acceptance under the belief that the authors will make the changes to their manuscript as they've described below.  I stand by my initial review as other reviewers also pointed out much of the same concerns; but I thank the authors for engaging & working toward what I believe will be a more clarified contribution.

=======

The authors latest response addressed many of my concerns, and I'd be happy to push for acceptance *contingent* on author commitment to make certain changes in the manuscript.

=======

I've increased my rating of this work from 3 to 4 based on the author response below.  Many more questions still remain (see my follow-up to author response).

=======

The authors provide a detailed documentation of their participation in the TREC-COVID challenge (Rounds 1-3).  Their system: (i) concatenates BioBERT sentence embeddings (Reimers and Gurevych 2019) with a standard inverted index over documents, and (ii) ranking documents using an averaging of BM25 scores and cosine similarities over the neural embeddings.  The system achieves moderate performance among automatic runs on the official TREC-COVID results.

Regarding the method, there is not sufficient Related Work provided to understand how this method fits within the broader literature  (has this been tried before -- if so, citation? is this similar to something else w/ some modification?  -- if so, citation & explanation/motivation of differences?  is this entirely novel?)

Regarding the relevance feedback (RF) baseline, why is this a sensible choice over (a) BioBERT sentence embeddings & cosine similarities without BM25, and (b) BM25 without the BioBERT sentence embeddings?  Comparing against the latter baselines would be able to help explain how/where the described system is performing well/not.

Regarding the evaluation, Table 1, while it is an accurate reporting of the official performance by this team, because the nature of the submissions differs between Rounds, it's hard to understand what learnings we can gain from this.  For example, RF appears for Round 1, but RF-RR for Round 2, and no baseline for Round 3.  Fusion appears for Round 3, but not Round 1 or Round 2.  NIRR appears for Round 2 and Round 3, but not Round 1.  It's really hard to see any meaningful patterns with results reported in this manner.  I believe the authors should've conducted separate controlled experiments to demonstrate efficacy of their method in a controlled setting, and supplement those results with the results from TREC-COVID, not base the entire paper on the TREC-COVID results.

Regarding the analysis in "Where does the model succeed or fail", the point about bias in TREC judgments is very interesting -- that is, their system returned a document at a high rank but was un-judged because no other system returned that document.  This could've been very interesting if the authors did a more thorough analysis -- how often does this happen? how big of an impact does this have on the performance scores (e.g. if these documents were judged, what would the scores look like?)   why might this model be behaving in this manner that's different from other systems?


Overall, this submission, while a perfectly fine detailed report about a team's submissions to a competition, does not propose a particularly powerful system or novel method (either of which would be great).  Nor does it perform enough careful experimentation/detailed analysis to glean any generalizable learnings from their experiences (which would also be great).

---

> ### Author Response · Authors · 2020-09-27
> **Detailed response to review**
>
> Thank you for your review and providing your perspective. We provide a detailed response to your review below.
>
> **"The authors provide a detailed documentation ...The system achieves moderate performance among automatic runs on the official TREC-COVID results."**
>
> While we agree that our system is not a 'LB winner' in terms of retrieval effectiveness, it was one of the top-performing systems (in NDCG@10, the metric used for the original leaderboard) in the first round of the shared task (thus, the only round with truly untuned automatic systems). Moreover, in our additional evaluations, we report a variant (Table 2) of the method, which would have achieved top NDCG scores in both rounds 1 and 2. This variant only differs from the officially submitted run with the BERT model used to produce the embeddings.
>
> **"Regarding the method, there is not sufficient Related Work provided to understand how this method fits within the broader literature (has this been tried before -- if so, citation? is this similar to something else w/ some modification? -- if so, citation & explanation/motivation of differences? is this entirely novel?)"**
>
> To the best of our knowledge, our approach is novel. Its conception (April 2020) was loosely inspired by [1], already cited in the paper.
> [1] https://ciir-publications.cs.umass.edu/pub/web/getpdf.php?id=1302
>
> **Regarding the relevance feedback (RF) baseline, why is this a sensible choice over (a) BioBERT sentence embeddings & cosine similarities without BM25, and (b) BM25 without the BioBERT sentence embeddings? Comparing against the latter baselines would be able to help explain how/where the described system is performing well/not.**
>
> The comparison against the baselines (a) and (b) requested by the reviewer is already presented for the Rd1 data in Table 3 and discussed in the manuscript. We will modify the manuscript to emphasise its presence. We are not quite sure how insensible is not including the baselines (a) and (b) within the limit of 3 runs per participant per round. While comparison against a RF baseline might not be the most informative, we believe it does answer a valid question - how well the proposed NIR model, which benefits from unsupervised pretraining and fine-tuning on an unrelated task, ranks against a classic RF feedback method (an interactive method proved to work well in biomedical literature retrieval).
>
> **Regarding the evaluation, Table 1...  it's hard to understand what learnings we can gain from this... I believe the authors should've conducted separate controlled experiments to demonstrate the efficacy of their method in a controlled setting, and supplement those results with the results from TREC-COVID, not base the entire paper on the TREC-COVID results.**
>
> Our intention was to compare NIR against a broad selection of different methods within the 3 runs limit. We agree that it would be ideal to provide a more thorough evaluation, but we submitted this work to a CovidNLP workshop, which, to the best of our understanding, is established as a forum to discuss ongoing research. The core of our submission is the novel NIR method, which yields promising results, as suggested by the evaluations we have carried out so far.
>
> **Regarding the analysis in "Where does the model succeed or fail", the point about bias in TREC judgments is very interesting -- that is, their system returned a document at a high rank but was un-judged because no other system returned that document. This could've been very interesting if the authors did a more thorough analysis -- how often does this happen? how big of an impact does this have on the performance scores (e.g. if these documents were judged, what would the scores look like?) why might this model be behaving in this manner that's different from other systems?**
>
> **Overall, this submission, while a perfectly fine detailed report about a team's submissions to a competition, does not propose a particularly powerful system or novel method (either of which would be great). Nor does it perform enough careful experimentation/detailed analysis to glean any generalizable learnings from their experiences (which would also be great).**
>
> As already mentioned, the method is novel. Moreover, the system actually did well in the first round of the evaluation and, without losing generality of the method, we present a variant which would've acheived top NDCG score in further round of the evaluation as well. While we agree that the paper would benefit from a more organised, systematic evaluation on multiple datasets, we feel we actually provided a broad range of meaningful comparisons within a short paper format, submitted to a thematic workshop.

---

> > ### Comment · AnonReviewer2 · 2020-09-29
> > **Response to author rebuttal**
> >
> > Thanks for the responses.
> >
> > *...it was one of the top-performing systems...in the first round of the shared task (thus, the only round with truly untuned automatic systems).*
> >
> > I agree that the NIR system performed well among automatic methods in Round 1, but this is a bit confusing.  This submitted paper discusses the NIR method in the context of its performance across Round 1-3, so necessarily we (reviewers) must assess the merits of the submission based on the reported Round 1-3 results.  And it is the case that in Rounds 2-3, NIR did not perform particularly strongly compared to other (automatic) systems in those rounds.
> >
> > *...we report a variant (Table 2) of the method, which would have achieved top NDCG scores in both rounds 1 and 2.*
> >
> > This is a fair point, and does improve my perception of this submitted work.
> >
> > *To the best of our knowledge, our approach is novel. Its conception (April 2020) was loosely inspired by [1], already cited in the paper.*
> >
> > This does not clarify things for me.  The proposed method as described in 4.1 looks like it uses BioBERT-NLI, which was trained by another party in the same way as done in Reimers and Gurevych.  So usage of this model to produce sentence embeddings is not inherently novel.  The citation of Zamani et al is odd because in that work, the sparse index is learned while it looks like in this submission, the sentence embeddings are concatenated with a traditional inverted index.  If anything, this is maybe more like the concatenated dense/sparse approach from Seo et al (https://www.aclweb.org/anthology/P19-1436.pdf), which was not in the literature review.
> >
> > If the approach is novel, then surely it warrants more discussion about how it is similar/different from these other works.  As 4.1 is written now, it's difficult to tell what the method is, let alone how it is novel or how it relates to these other works.
> >
> > *The comparison against the baselines (a) and (b) requested by the reviewer is already presented for the Rd1 data in Table 3 and discussed in the manuscript. We will modify the manuscript to emphasise its presence.*
> >
> > Thank you.  I misunderstood what was happening in Table 3.  The discussion & Table are currently a bit confusing (text mentions "cosine scores" while Table 3 doesn't say "cosine scores", but only "neural";  Table 3 shows P@5 but text never discusses it.
> >
> > Yet, now assessing these results again, more questions actually arise than are addressed.  It seems removing the neural component from the NIR model results in an improvement, then it does seem the NIR method that uses neural embeddings is actually worse than just running BM25.  Yes, Table 2 does show improvements as a result of swapping out different choice of BERT model, but it's still not entirely clear what's happening.  The best BERT model is ClinicalCovid-NLI, but both Covid-NLI and SciBERT-NLI are substantially worse than just base NIR.
> >
> > What are these models?  I don't know what ClinicalCovid-NLI, Covid-NLI, and SciBERT-NLI are (there are no citations, URLs, descriptions, etc.)  Why did "NLI" work for one of them but not the other?  Are "ClinicalCovid", "Covid", and "SciBERT" models (prior to NLI) very different, or what evidence do you have that these experimental results are not an artifact of high variance/noise and more a result of some underlying phenomenon?  Can you discuss this?
> >
> > *We are not quite sure how insensible is not including the baselines (a) and (b) within the limit of 3 runs per participant per round.*
> >
> > The 3 runs per participant is an odd thing to bring up.  From results in Table 2 and Table 3, it doesn't seem like TREC-COVID run limits have prevented you from conducting relevant experiments?
> >
> > *While comparison against a RF baseline might not be the most informative, we believe it does answer a valid question - how well the proposed NIR model, which benefits from unsupervised pretraining and fine-tuning on an unrelated task, ranks against a classic RF feedback method (an interactive method proved to work well in biomedical literature retrieval).*
> >
> > This is a fair point, and does improve my perception of this submitted work.
> >
> > *Overall*
> >
> > My concern here is, in the current submission, the experiments have not been thoroughly motivated, documented or discussed for me to know what I should be taking away from this work.  I'm left with more questions than were answered
> >
> > - What are models in Table 2?
> > - What's the takeaway from Table 2 & how do you know it's not due to noise?
> > - Why switch from NIR_{cls} to NIR_{avg}?
> > - No discussion of some experiments (i.e. title, abs, etc.) in Table 3
> > - Regarding Figure 2, when you say the neural component alleviates effect of pure word matching, can you verify this from NIR_{avg} - neural from Table 2?
> > - Again, how big a deal is the irrelevant rank 1 document for Topic 3 in affecting overall scores?
> > - What exactly are the modeling contributions, and how is it similar/different from previous work, precisely?
> > - Are NIRR/Fusion relevant?

---

> > > ### Author Response · Authors · 2020-09-29
> > > **Response to questions**
> > >
> > > Thank you for providing a response and your interest.
> > >
> > > **"This does not clarify things for me. The proposed method as described in 4.1 looks like it uses BioBERT-NLI... So usage of this model to produce sentence embeddings is not inherently novel. The citation of Zamani et al is odd because in that work, the sparse index is learned ... the sentence embeddings are concatenated with a traditional inverted index. If anything, this is maybe more like the concatenated dense/sparse approach from Seo et al ... which was not in the literature review... how it is novel or how it relates to these other works."**
> > >
> > > We agree that our approach is similar in using both sparse and dense documents representations for a ranking problem.
> > > However, our method is different in virtually every other aspect.
> > > * our embedding model is only finetuned for the NLI task (so, on a task somewhat unrelated to literature retrieval).
> > > * at inference, our strategy caters to embedding scientific literature (aggregated sentence embeddings to represent the abstract; a sentence embedding of the title), not individual sentences
> > > * we propose a scoring function, which combines the results of traditional inverted index-based retrieval (BM25)
> > >
> > > **" Thank you. I misunderstood what was happening in Table 3. The discussion & Table are currently a bit confusing (text mentions "cosine scores" while Table 3 doesn't say "cosine scores", but only "neural"; Table 3 shows P@5 but text never discusses it."**
> > >
> > > We will expand the caption to clarify the table’s contents.
> > >
> > > **"It seems removing the neural component from the NIR model results in an improvement, then it does seem the NIR method that uses neural embeddings is actually worse than just running BM25... The best BERT model is ClinicalCovid-NLI...**"
> > >
> > > Yes, it turns out that NIR_{avg} does worse than BM25 in terms of ranking but in terms of precision, it performs better. There is a trade-off. However, we found ClinicalCovid-NLI (which we use in subsequent rounds) is better than BM25 in both.
> > >
> > > **"What are these models? I don't know what ClinicalCovid-NLI, Covid-NLI, and SciBERT-NLI are (there are no citations, URLs, descriptions, etc.)"**
> > >
> > > We will add a description of those.
> > >
> > > **"Why did "NLI" work for one of them but not the other? Are "ClinicalCovid", "Covid", and "SciBERT" models... what evidence do you have that these experimental results are not an artifact of high variance/noise and more a result of some underlying phenomenon? Can you discuss this? What's the takeaway from Table 2 & how do you know it's not due to noise?**"
> > >
> > > NLI refers to finetuning base models (such as CovidBERT, SciBERT, BiOBERT) with a cosine regression objective over NLI datasets within the siamese network framework proposed by Guryevich et al. The differences between the models stem mainly from the datasets were pretrained on. Because our objective is to use these models to embed biomedical documents, we chose models that were pretrained with biomedical documents. ClinicalCovid works better than BioBERT and SciBERT (proven effective on other biomedical tasks) because it has been trained on a sizeable corpus related to Covid-19. BioBERT/SciBERT models were created before the COVID started and thus is not in its pretraining data (which is why it performs worse). COVID-NLI is an interesting case, as this was pretrained relatively early on (when there weren't as many CORD-19 documents) which is why it's weaker than ClinicalCovidBERT (which was trained later).
> > >
> > > The point of the table was to compare against other embedding models and its effect on the retrieval performance of the NIR model.
> > >
> > > **"Why switch from NIR_{cls} to NIR_{avg}?**"
> > >
> > > When inspecting the attention spectrum of the final layers of the embedding models. We found that attention was not pooled onto the classification token which we previously assumed, changing from CLS to AVG pooling improved performance from 0.588 (reported in competition; Table 1 Round 1) to 0.614 (Table 2).
> > >
> > > **Regarding Figure 2, when you say the neural component alleviates effect of pure word matching, can you verify this from NIR_{avg} - neural from Table 2?**"
> > >
> > > In terms of precision, we can say that the model retrieved more relevant documents, but the ranking performance (NDCG) suffered as a result. Queries 24-25 are queries which we performed better, mainly because the best run retrieved were only related to diabetes generally rather than inferring diabetes and COVID-19 together (our run).
> > >
> > > **"Again, how big a deal is the irrelevant rank 1 document for Topic 3 in affecting overall scores?**"
> > >
> > >  This instance is not affecting the overall scores in isolation, we use it more like an illustrative example.
> > >
> > > **"Are NIRR/Fusion relevant?**"
> > >
> > > Not particularly, those were just some of the other ideas we’ve tried. These are only reported for completeness, but we agree that the paper could probably do without them and be replaced with more explanation of the other ideas mentioned.

---

> > > > ### Comment · AnonReviewer2 · 2020-09-30
> > > > **Willing to greatly increase my review score, contingent on revisions**
> > > >
> > > > Regarding the authors response to my question around clarifying the novelty of their method, I am not satisfied with the response.  I don't feel the bullet list of slight modifications to existing methods are compelling enough as actual contributions to claim a 'novel method'.
> > > >
> > > > That being said, I read the authors response to Reviewer 3's question about what the main contribution of the paper is.  The authors response was:
> > > >
> > > > *The main contribution of the paper is mainly to show that neural representations (that are universal sentence embeddings) can be used as part of a neural index. Representations from contemporary transformer models typically cannot use cosine similarity, however, this can be addressed by training on a cosine similarity objective from [1]. In our method, is no 'direct' re-ranking step as in traditional systems where BM25 is used to first perform a search and a neural model then reranks documents in a pipeline manner, the system we propose is end-to-end which is a novel contribution and was among the top-ranking systems.*
> > > >
> > > > This is a very good response & clarifies a lot of what I was confused about in this submission.  And it adequately addresses my own questions about the novelty of their method.  If the authors include this in their manuscript, it certainly improves my perception of the work.
> > > >
> > > > --------
> > > >
> > > > Regarding authors response to my question about the other BERT models (BioBERT, SciBERT, CovidBERT, etc.) and their NLI variants, this was a very good response.  And I think the authors explanation about how ClinicalCovid has a lot more COVID19 papers involved is very important.  If the authors will include this explanation in their manuscript, it would also improve my perception of the work.
> > > >
> > > > --------
> > > >
> > > > Finally, reading some of the other reviews, I definitely get the feeling that the current submission is hard to follow because the authors have attempted to structure the paper around their participation in TREC-COVID.  But I think this work has some valuable content (see aforementioned point about their main contribution, and their interesting experiments/interpretation of results with different BERT models)  that is hard to see because there's all this other extra stuff that's confusing (e.g. Reviewer 2 was also confused by the presence of NIRR).
> > > >
> > > > I think if the authors are willing to prune some of this auxiliary details from the paper, and focus entirely on the things that they're trying to contribute & the interesting findings they observed, then I'd be very happy with this submission.
> > > >
> > > > Can the authors respond to describe a bit about what their intentions are with the manuscript with respect to this?
> > > >
> > > > --------
> > > >
> > > > Finally, this is a minor point, but I didn't enjoy the "illustrative example" around the rank 1 document for Topic 3.   Given that it didn't actually affect the ranking, I felt this was a misleading cherry-picked example.  Hopefully this can be removed

---

> > > > > ### Author Response · Authors · 2020-10-01
> > > > > **Intentions for revision**
> > > > >
> > > > > We thank you for the detailed comments and continued dialogue. We believe they will help greatly in improving our submission. Our revision plans for the manuscript, acceptance aside, are as follows:
> > > > >
> > > > > 1. We will add a clear contribution statement based on our responses to the reviewers:
> > > > > The main contribution of the paper is a model leveraging neural representation (that are universal sentence embeddings) used as part of a neural index. Representations from contemporary transformer models typically cannot be used for cosine similarity, however, this can be addressed by training on a cosine similarity objective from [1]. In our method, there is no 'direct' re-ranking step as in traditional systems where BM25 is used to first perform a search and a neural model then reranks documents in a pipeline manner, the system we propose is end-to-end, which is a novel contribution. The proposed system was among the top-ranking systems in the initial rounds of the TREC COVID Challenge.
> > > > > 2. We will add a clarification on the nature of BERT-based models used in our experiments (based on our earlier response)
> > > > > 3. We will remove the experiments that are not relevant to the main contribution: NIRR, RF_RR.
> > > > > 4. In round-by-round format we will report NIR_CLS, NIR_avg with basic and optimal BERT model, BM25 and RF as baselines, TREC best for reference (so, 7 results per round, consistent between round). We will include information that NIR_CLS is our ‘official’ runs evaluated within TREC shared task, but we will not relate our contributions to the shared task model-by-model.
> > > > > 5. We will clarify the discussion of the qualitative examples. I.e., the discussion of top document for topic 3 actually presents a false positive (so, the opposite of cherry-picking), where our intention was to show that BM25 can still ‘overcome’ the neural component completely if the words match. We will also report the proportion of unjudged documents @10 when discussing the ‘unjudged’ example to provide quantitative perspective.

---

### Official Review · AnonReviewer3 · 2020-09-29
**A detailed report of experiments with mediocre contribution.**

**Rating:** 5
**Confidence:** 3

**Review:**

This paper proposes a simple and effective model for the COVID-19 challenge that combines similarity scores computed from neural representations and traditional inverted index scores (BM25). The details are extensively provided however the organization can be improved, as it's unclear what are the motivations for technical choices and baseline choices.

1. My biggest concern is that the contribution of this paper is not clear. It seems the biggest discovery is that BM25 score is very important for the model (and potentially for other models that participate in this challenge). While this can be a contribution, but the argument sounds weak to me. Isn't this a widely used tool even outside of this challenge? Furthermore, in the ablation study, when taking out BM25 components, the performance dropped to a point of unusable. This questions whether the neural part of this model is effective. It looks like the BM25 is doing heavy-lifting there.

2. What does the NIRR model add to this paper? It seems it performed worse than NIR but the paper does not offer any analysis.

3. It seems another contribution buried in the technical details is that "the NIR model...performs on par with the RF run with neural reranking, without the need for tasks specific training data." So the proposed method is actually unsupervised?  I am confused. Did it use round 1 judgement during round 2? and in round 3?

---

> ### Author Response · Authors · 2020-09-29
> **Contributions of the paper and insights**
>
> Thank you for your review, we provide a detailed response to clarify your questions below.
>
> **What is the main contribution of paper?**
>
> The main contribution of the paper is mainly to show that neural representations (that are universal sentence embeddings) can be used as part of a neural index. Representations from contemporary transformer models typically cannot use cosine similarity, however, this can be addressed by training on a cosine similarity objective from [1]. In our method, is no 'direct' re-ranking step as in traditional systems where BM25 is used to first perform a search and a neural model then reranks documents in a pipeline manner, the system we propose is end-to-end which is a novel contribution and was among the top-ranking systems.
>
> [1] Nils Reimers and Iryna Gurevych. 2019. SentenceBERT: Sentence embeddings using Siamese BERT networks. In EMNLP, pages 3982–3992, Hong Kong, China.
>
> **While this can be a contribution, but the argument sounds weak to me. Isn't this a widely used tool even outside of this challenge? Furthermore, in the ablation study, when taking out BM25 components, the performance dropped to a point of unusable. This questions whether the neural part of this model is effective. It looks like the BM25 is doing heavy-lifting there.**
>
> While we use BM25 as a backbone to our system, the neural component adds equal contribution to the score. The main purpose is that the cosine similarity measure is not directly usable for ranking (see discussion) which is why we see the performance drop significantly in Table 3 on the removal of BM25. However, from Table 2, we can see that changing the embedding model (e.g. ClinicalCovid-NLI which can better embed the CORD-19 documents) allows us to beat the best automatic models.
>
> For clarity, the neural score + BM25 score together is better than the bm25 score alone especially if we use the ClinicalCovid-NLI model (as NIR - neural is BM25 score) and is also better than the best automatic runs for rounds 1 & 2 which use traditional keywords approaches such as BM25. But we cannot use neural score alone, as it is not a suitable measure for ranking or relevance.
>
> Our main discovery was not intended to show that BM25 is the backbone of the model. Instead, our intention was to show that with the right formulation, we can produce embeddings at index-time (rather than at query-time) and use these in tandem with a traditional scorer with a ranking objective (BM25) which makes the resulting model harness the benefit of both contextual representations and ranking. We also then discuss the successes and failures of the model.
>
> **What does the NIRR model add to this paper? It seems it performed worse than NIR but the paper does not offer any analysis.**
>
> Our intention was to compare NIR against a broad selection of different methods within the 3 runs limit. We agree that it would be ideal to provide a more thorough evaluation, but we submitted this work to a CovidNLP workshop, which, to the best of our understanding, is established as a forum to discuss ongoing research. The core of our submission is the novel NIR method, which yields promising results, as suggested by the evaluations we have carried out so far.
>
> **It seems another contribution buried in the technical details is that "the NIR model...performs on par with the RF run with neural reranking, without the need for tasks specific training data." So the proposed method is actually unsupervised? I am confused. Did it use round 1 judgment during round 2? and in round 3?**
>
> Our proposed NIR method is completely unsupervised and performed on par with a relevance feedback model that had annotations. In round 1, The RF model used hand-annotated documents (which was done internally and has a direct link in the paper), and for round 2, the RF model used relevance judgments from round 1 provided by the organizers.
>
> We emphasize that our NIR method is fully automatic (unsupervised, using a universal sentence embedding model for biomedical text), and did not rely on any relevance judgments (which is why it has no concept of ranking).

---

### Official Review · AnonReviewer1 · 2020-09-30
**System paper for TREC-COVID**

**Rating:** 4
**Confidence:** 3

**Review:**

This paper describes an approach to the TREC COVID Search challenge. The proposed method combines an inverted index score using BM25 and a cosine similarity score based on neural representations derived from BioBERT-NLI. The paper's main contribution is the proposal of a hybrid system that can perform document retrieval without separate search and re-ranking steps using a pre-trained neural indexer.

1. A large concern about the paper is its incoherency in supporting its main contribution. The paper denotes NIR_{AVG} as its proposed method, but the model is neither the official submitted model (NIR_{[CLS]}) nor the best performing model in the additional run (ClinicalCovid-NLI). What's the rationale behind this choice?
2. It would have been nice to see quantitative analysis of models' performance regarding "Where does the model succeed or fail?"  The authors provide a qualitative description of a case where the model had lost points due to unjudged documents, but the explanation's qualitative nature leaves many questions open. Is it always the case that the model's low NDCG@10 scores in different topics due to this phenomenon? How does it improve as the base model changes to a stronger model?

Additional comments:
- The paper does not describe why certain documents were unjudged. One can guess that it's related to other teams' submissions, but it would be better to have the exact criteria described in the document.
- Why doesn't Table 2 have the numbers for Round 3?

---

> ### Author Response · Authors · 2020-10-01
> **Response to reviewer**
>
> Thank you for your review.
>
> We aim to overhaul the paper from the responses from your review and the other reviewers. We posted a response to reviewer 2 but will repost our intentions here for visibility.
>
>  Our revision plans for the manuscript, acceptance aside, are as follows:
> 1. We will add a clear contribution statement, based on our response to Reviewer 3.
> The main contribution of the paper is a model leveraging neural representations (that are universal sentence embeddings) used as part of a neural index. Representations from contemporary transformer models typically cannot be used for cosine similarity, however, this can be addressed by training on a cosine similarity objective from [1]. In our method, there is no 'direct' re-ranking step as in traditional systems where BM25 is used to first perform a search and a neural model then reranks documents in a pipeline manner, the system we propose is end-to-end, which is a novel contribution. The proposed system was among the top-ranking systems in the initial rounds of the TREC COVID Challenge.
> 2. We will add a clarification on the nature of BERT-based models used in our experiments (based on our earlier response)
> 3. We will remove the experiments that are not relevant to the main contribution: NIRR, RF_RR.
> 4. In round-by-round format we will report NIR_CLS, NIR_avg with basic and optimal BERT model, BM25 and RF as baselines, TREC best for reference (so, 7 results per round, consistent between round). We will include information that NIR_CLS are our ‘official’ runs evaluated within TREC shared task, but we will not relate our contributions to the shared task model-by-model.
> 5. We will clarify the discussion of the qualitative examples. I.e., the discussion of top document for topic 3 actually presents a false positive (so, the opposite of cherry-picking), where our intention was to show that BM25 can still ‘overcome’ the neural component completely if the words match. We will also report the proportion of unjudged documents @10 when discussing the ‘unjudged’ example to provide quantitative perspective.
>
> **The paper does not describe why certain papers are unjudged..**
>
> The judgment pool is a collection of runs (chosen by the organizers) to be manually annotated by biomedical experts. Our NIR model was not part of the judgment pool for Round 1. We will include this information along (and subsequent discussions pertaining to our results) in a revised manuscript.